# Isothioureas, Ureas, and Their *N*-Methyl Amides from 2-Aminobenzothiazole and Chiral Amino Acids

**DOI:** 10.3390/molecules24183391

**Published:** 2019-09-18

**Authors:** Itzia I. Padilla-Martínez, José Miguel González-Encarnación, Efrén V. García-Báez, Alejandro Cruz, Ángel Andrés Ramos-Organillo

**Affiliations:** 1Instituto Politécnico Nacional-UPIBI, Laboratorio de Química Supramolecular y Nanociencias, Av. Acueducto s/n, Barrio la Laguna Ticomán 07340, México City, Mexico; ipadillamar@ipn.mx (I.I.P.-M.); jos3mi6uelglez@gmail.com (J.M.G.-E.); efren1003@yahoo.com.mx (E.V.G.-B.); 2Facultad de Ciencias Químicas, Universidad de Colima, Km 9 Carr. Colima-Coquimatlán, Coquimatlán 28400, Colima, Mexico; aaramos@ucol.mx

**Keywords:** 2-aminobenzothiazole, 2-dithiomethylcarboimidatebenzothiazole, α-amino-acids, S-methyl-isothioureas, urea-carboxylate methyl esters and urea *N*-methyl amides

## Abstract

In this investigation, the reaction of 2-dithiomethylcarboimidatebenzothiazole with a series of six chiral amino-acids was studied. The reaction proceeds through the isolable sodium salt of SMe-isothiourea carboxylates as intermediates, whose reaction with methyl iodide in stirring DMF as solvent affords SMe-isothiourea methyl esters. The presence of water in the reaction leads to the corresponding urea carboxylates as isolable intermediates, whose methyl esters were obtained. Finally, the urea *N*-methyl amide derivatives were isolated when SMe-isothiourea or urea methyl esters were reacted with methylamine in the presence of water. The structures of synthesized compounds were established by ^1^H and ^13^C nuclear magnetic resonance and the structures of SMe-isothiourea methyl esters derived from (*l*)-glycine, (*l*)-alanine, (*l*)-phenylglycine, and (*l*)-leucine, by X-ray diffraction analysis. This methodology allows to functionalize 2-aminobenzothiazole with SMe-isothiourea, urea, and methylamide groups derived from chiral amino acids to get benzothiazole derivatives containing coordination sites and hydrogen bonding groups. Further research on the biological activities of some of these derivatives is ongoing.

## 1. Introduction

Benzothiazole is an aromatic bicyclic ring system that consists of a thiazole ring fused with a benzene ring. The benzothiazole moiety is small but a very interesting compound with wide biological activities. At the beginning of the 1970s, it was found that benzothiazole derivatives possessed pharmacological antiviral [1,2,3], antibacterial [3,4,5,6], antimicrobial [7,8,9], fungicidal [10,11], antiallergic [12,13,14], antidiabetic [15,16,17], antitumoral [18,19,20,21], anti-inflammatory [22,23] and anthelmintic [24,25,26] activities.

A continuous interest in this class of compounds follows nowadays and numerous efforts to synthesize new biologically active heterocyclic compounds derived from the benzothiazole moiety have been made in the last 50 years. Several of these derivatives were found to possess anticonvulsant [27,28,29] and antioxidant [30,31,32] activities. In this context, some results related to molecules containing benzothiazole nuclei in medicinal chemistry have been summarized elsewhere [33,34].

On the other hand, guanidines are an important class of compounds that are found widely throughout nature and as biologically active synthetic compounds, and several uses in organic chemistry are known [35,36,37,38]. For example, the natural amino acid arginine has a guanidine group as side chain, whilst cimetidine, a synthetic guanidine-derived compound, was the first drug used to treat peptic ulcers.

Typically, the reaction of amines with thioureas [39] or isothioureas [40,41,42] is the most commonly used method to obtain guanidines. Particularly, the isothiourea group has been bonded to a solid phase as precursor of guanidines [43]. In the last decade, it has been recognized that isothioureas can also serve as remarkably potent inhibitors for a range of enzymatic systems [44,45,46]. On the other hand, inhibition of nitric oxide synthase (NOS) has led to their use in the treatment of a range of life-threatening conditions, including septic shock, acute kidney failure, and rejection after transplantation surgery [47].

In this sense, we report a synthetic way to prepare symmetrical and nonsymmetrical guanidines **4** derived from 2-aminobenzothiazole **1** by the reaction of 2-dithiomethylcarboimidatebenzothiazole **2** with primary amines in refluxing ethanol [48]. The reaction proceeds through the formation of SMe-isothioureas **3**, as intermediates, and the displacement of two MeSH molecules [49], Scheme 1.

In this contribution, we applied this methodology in the reaction of **2** with a series of chiral amino acids to form the corresponding SMe-isothiourea carboxylates **5**, which were further transformed into the corresponding isourea carboxylates **6**, methyl ester derivatives **8**–**10** and **12**, and isourea amides **11**, and **13**. This methodology allows the functionalization of 2-aminobenzothiazole to introduce groups such as isothioureas, isoureas ureas, amides, and guanidines derived from chiral amino acids. These functional groups give properties such as coordinating sites, hydrogen bonding interactions, and water solubility, among others, required for the interaction with biomolecules. The reactions were carried out without modification of the chiral center configuration and all compounds were optically pure isolated as shown in the analyzed X-ray structures of compounds **8b**,**c** and **9f**.

## 2. Results and Discussion

### 2.1. Reactions and Characterization

Six amino acids **Aa**–**f** were tested: glycine (R = H, **a**), (*l*)-alanine (R = Me, **b**), (*l*)-phenylglycine (R = Ph, **c**), (*l*)-phenylalanine (R = Bn, **d**), (*l*)-valine (R = ^i^Pr, **e**), and (*l*)-leucine (R = ^i^Bu, **f**), represented as the zwitterions **Ba**–**f**. Each amino acid was transformed in situ into the corresponding sodium carboxylate **Ca**–**f** by reaction with one molar equivalent of sodium hydroxide in stirring ethanol for 2 h at room temperature, Scheme 2.

The reaction of one molar equivalent of 2-dithiomethylcarboimidatebenzothiazole **2** with the corresponding amino-acid carboxylates **Ca**–**f** at refluxing ethanol for eight hours was carried out. In these conditions, the reaction proceeds by nucleophilic attack of the amino group of the amino-carboxylates **C** to the carbonimidothioate group of compound **2**, with elimination of thiomethanol gas to afford the corresponding SMe-isothiourea-carboxylates **5a–f** in 40**–**86% yields. The ^1^H NMR spectroscopic data of compounds **5a**–**f** are listed in Appendix A. A singlet for the SMe group is found in the 2.35–2.44 ppm range, whose integration area are in a 3:4 proportion in relation to the aromatic hydrogen atoms. The chemical shift of the NH group to high frequency (10.6–11.4 ppm) is explained due to a hydrogen bonding interaction with the nitrogen atom of the benzothiazole moiety, as depicted in Scheme 3. On the other hand, the ^13^C NMR data, listed in Appendix A, show characteristic signals from 13.9 to 14.1 and 169.7 to 171.6 ppm ranges for the SMe and OC=O groups, respectively.

In the case of the reaction of compound **2** with valine- (**Ce**, R = ^i^Pr) or leucine- (**Cf**, R = ^i^Bu) carboxylates, an insoluble yellowish solid appeared in the reaction mixture, which was identified as compound **7**. The ^1^H NMR spectrum shows two singlets at 2.6 ppm (SMe) and 3.9 (NMe) each in a 3:4 proportion with respect to the aromatic hydrogen atoms. In the ^13^C NMR spectrum, the signals for SMe, NMe, and the thiocarbonyl group at 18.6, 33.7, and 208.7 ppm, respectively, appeared. To explain these results, a sigmatropic rearrangement of compound **2** to form compound **7** in 15% yield is proposed, as depicted in Scheme 4, in agreement with ^1^H and ^13^C NMR data.

Isorea carboxylate compound **6c** was isolated as byproduct in 20% yield, from the remaining mother liquors of **5c**. No signal for the SMe group was present in the NMR spectra of compound **6c**, but two interchangeable protons with deuterium were observed at 11.8 ppm, corresponding to the isourea OH, and at 8.3 ppm, attributed to the urea NH. ^1^H and ^13^C NMR spectroscopic data of compound **6c** are listed in Appendix A, respectively. The substitution of the remaining SMe group in compound **5c** by one molecule of water afforded compound **6c** as one of the two possible tautomers, Appendix A.

To improve the yields of the sodium salt of SMe-isotiourea carboxylates **5b**–**f**, the reactions were carried out using anhydrous ethanol and stirring for 4 days at room temperature to avoid hydrolysis. In these conditions, the reaction proceeds more slowly to afford the corresponding SMe-isothiourea-carboxylates **5b**–**f** in 62–95% yields. On the other hand, the complete hydrolysis of SMe-isotiourea carboxylates **5b**–**f** in a refluxing mixture of ethanol:water 1:1 was carried out. In these conditions, the second thiomethanol gas molecule was eliminated to afford the corresponding isourea carboxylates **6b**–**e** as the only products in 65–75% yields. The hydrolysis of compound **5a** required more drastic conditions such as refluxing in DMF/H_2_O mixtures to obtain **6a** in 40% yield.

The corresponding SMe-isothiourea carboxylate methyl esters **8a**–**d** were obtained in 56–83% yields after the methylation of carboxylates **5a**–**d** with one molar equivalent of methyl iodide in DMF as solvent, whose ^1^H and ^13^C NMR chemical shifts are listed in Appendix A, respectively. In general, their spectra are very similar compared to the corresponding carboxylates **5a**–**d**, except for the OMe group signals which are in the 3.75–3.83 and 45–53 ppm ranges in ^1^H and ^13^C NMR spectra, respectively. The ^13^C NMR data of esters **8c**,**d** in CDCl_3_ show broad signals for C2, C9, and C11, suggesting that the usually fast proton exchange between N3 and N12 through tautomeric equilibria becomes slower because of the steric effects of the phenyl and benzyl moieties from the amino-acid residue.

The use of one equivalent of iodomethane in the reaction of SMe-isothiourea-carboxylates **5e** or **5f**, afforded the respective methyl esters **8e** (49% yield) or **8f** (51% yield) in mixture with the corresponding N3-Me methyl esters **9e** (8% yield) or **9f** (25% yield) and their hydroiodides **9e**∙HI or **9f**∙HI. Compound **5f** was reacted with two molar equivalents of CH_3_I; however, compounds **8f** and **9f**∙HI remained. This last compound precipitated from the reaction mixture and was separated for further analysis. A ^1^H and ^13^C chemical shifts comparison between compounds **9e**, **9f**∙HI, and **9f** is depicted in Figure 1. The characteristic ^1^H (^13^C) NMR signals of **9f**∙HI, are the SMe group at δ 3.1 (18.3), N–Me at 3.8 (56.5), and N–H at 9.1 ppm. The high frequency shift of the NH suggests a hydrogen bonding interaction with the sulfur atom and/or with the carbonyl oxygen atom, Figure 1. The nitrogen atom of the N–Me group on the benzothiazole produces an electronic effect on C4 of the aromatic ring, shifting it to low frequencies: 114.1 ppm for **9f**∙HI and ≈ 110 ppm for **9e** and **9f**.

The isourea-carboxylates **6a**–**d** can also be methylated to afford the isourea methyl esters **10a**–**d** in 30–90% yields, Scheme 3. However, the methylation reaction of the sodium salts of isourea-carboxylates **6e** or **6f** afforded the corresponding methyl esters **10e** (66%) or **10f** (54%) in mixture with their N–Me esters **12e** (24%) or **12f** (21%). ^1^H and ^13^C NMR data of compounds **10a**–**f** are listed in Appendix A, respectively, and those of urea-methyl esters **12e** and **12f** are depicted in Figure 2.

Isourea carboxylate methyl esters **10a**–**f** or **12e**,**f** were reacted with methylamine to afford their corresponding isourea amides **11a**–**f** or urea-amides **13e**,**f** in 60–97% yield or 20% and 46% yields, respectively. The ^1^H and ^13^C NMR spectra of compounds **11a**–**f** are listed in Appendix A and those of compounds **13e**,**f** are depicted in Figure 2. The ^1^H NMR spectrum of isourea-amides **11a**–**f** show three deuterium labile hydrogen atoms in the 8.4–10.9, 7.0–10.6, and 7.1–8.1 ppm ranges, as well as the characteristic doublet in 2.5–3.0 ppm range for the NHMe group. The C4 NMR frequencies were found at approximately 110 ppm in both NMe esters **12e**,**f** and their amides **13e**,**f**. The 10 ppm shift to low frequencies compared with their NH analogues **10a**–**f** and **11a**–**f** was due to the electronic effect of the NMe group on C4. In compounds **13e** and **13f**, the urea NH appears as a doublet at 5.9 (^3^*J* = 9.3 Hz) and 5.7 ppm (^3^*J* = 8.2 Hz); and the amide NH appears at lower frequency as a quartet at 6.6 (^3^*J* = 4.4 Hz) and 6.4 ppm (^3^*J* = 4.7 Hz), respectively.

Isothiourea carboxylate methyl esters **8a**–**f** contain both SMe and OMe groups, which are susceptible to substitution with nucleophiles such as methylamine. The reaction of compounds **8a** with one molar equivalent of methylamine produces a complex mixture of several methylated compounds. However, in the presence of an excess of methylamine, the isourea-amide compound **11a** precipitated as a white solid. In these conditions, the SMe group was substituted because of the formation of MeNH_3_OH in aqueous medium. The last procedure was also used with the urea methyl esters **8b**–**f** and **9e**,**f** to obtain compounds **11b**–**f** and **13e**,**f** in 45–60% yields.

### 2.2. Molecular Structure of Compounds ***8a**–**c*** and ***9f***

The SMe-isothiourea methyl esters (*S*)-**8a**–**c** and (*S*)-**9f** were purified by crystallization from ethanol and suitable crystals for X-ray diffraction analysis were isolated. The molecular structures of compounds **8a** and **8c**, displayed in Figure 3 and Figure 4, show that the N12H is engaged in intramolecular three-centered hydrogen bonding interaction with the benzothiazole nitrogen and carbonyl oxygen atoms. The distances and angles associated with this N3···H12···O14 interaction are N12H···N3 = 2.01 Å, 132° (**8a**) and 2.03 Å, 132° (**8c**); N12H···O14 = 2.31 Å, 107° (**8a**) and 2.21 Å, 111° (**8c**), forming the corresponding adjacent six (*S6*) and five (*S5*)-membered rings. This hydrogen bonding interaction fixes the stereochemistry of the imine N10–C11 bond and only the (*E*) isomer of **8a** and **8c** was produced. In addition, the lateral side chain is almost in the same plane of the benzothiazole, including the carbon atoms of both OMe and SMe groups. In general, the SMe group is the most deviated from the mean plane [N(10)-C(11)-S(23)-C(24) −6.18(1)°] compared with the OMe group [O(14)-C(14)-O(15)-C(16) 0.25(1)°]. In the structure of compound (*S*)-**8b**, displayed in Figure 5, only the intramolecular hydrogen bonding interaction with benzothiazole nitrogen atom was observed, N12H···N3 (2.06 Å, 133°), forming the corresponding six membered ring (*S6*). Therefore, the isothiourea group is in the same plane of the benzothiazole, including the chiral carbon atom, N(10)-C(11)-N(12)-C(13) −178.71(1)° and S(23)-C(11)-N(12)-C(13) 0.16(1)°. In this case, the carbon atom of the SMe group is deviated from the benzothiazole ring planes, N(10)-C(11)-S(23)-C(24) = 5.19(1)° and the carboxylate group is almost perpendicular to the plane of the molecule C(14)-C(13)-N(12)-C(11) = −85.82(1). 

The structure of compound **9f** is displayed in Figure 6. Two intramolecular noncovalent bonding interactions were observed, one of them is that of N14 with benzothiazole sulfur atom, S···N14 (2.687 Å), the other of S12 with carbonyl carbon atom S12···C16 (3.123 Å), forming in both cases a five (S5)-membered ring. The first interaction causes both exocyclic nitrogen atoms to be almost in the same plane of the benzothiazole rings, including the chiral carbon and methylene carbon atoms of the isobutyl group. The SMe group is approximately 10° deviated from the mean plane, S12-C11-N10-C2 −169.8(4) compared with a small deviation of N10 [N10-C2-N3-C23 −3.6(7)°] and C11 [S1-C2-N10-C11 2.1(7)°].The second interaction maintain the C=O and the isopropyl groups to be opposite each other deviated from the mean plane C16-C15-N14-C11 −70.1(5) and N14-C15-C19-C20 −66.1(6), respectively. Intermediate bond lengths values between single and double character for C2-N3 [1.330(5) Å], C2-N10 [1.309(6) Å], N10-C11 [1.369(5) Å], whereas double bond for C11-N14 [1.283(6) Å] and single bond for N14-C15 [1.461(6) Å] were observed.

## 3. Materials and Methods

Melting points were measured on an IA 9100 apparatus (Electrothermal, Staffordshire, UK) and are uncorrected. IR spectra were recorded using a 3100 FT-IR Excalibur Series spectrophotometer (Varian, Randolph, MA, USA) equipped with an ATR system. Mass spectra were obtained in a 3900-GC/MS system (Varian, Palo Alto, CA, USA) with an electron ionization mode. Elemental analyses (EA) were performed on a 2400 elemental analyzer (Perkin-Elmer, Waltham, MA, USA). ^1^H- and ^13^C-NMR spectra were recorded on a Varian Mercury 300 (^1^H, 300.08; ^13^C, 75.46 MHz) instrument in DMSO-d_6_ solutions for compounds 5a–f and 6a–f, and CDCl_3_ in solutions for compounds **8a**–**f**, **9a**–**f**, **10a**–**f**, **11a**–**f**, **12e**,**f**, and **13e**,**f**; SiMe_4_ was used as the internal reference. Chemical shifts are in ppm and *^n^**J*(H-H) in hertz.

Crystals suitable for X-ray analysis of **8a**, **8b**, **8c**, and **9f** were obtained after solvent evaporation from saturated ethanol solutions. Single-crystal X-ray diffraction data were recorded on a D8 Quest CMOS (Bruker, Karlsruhe, Germany) or Nonius Kappa (Rotterdam, the Netherlands) area detector diffractometers with Mo K α radiation, λ = 0.71073 Å. A table listing the crystallographic data is provided as Appendix A. The structures were solved by direct methods using SHELXS97 [50] program of WinGX package [51]. The final refinement was performed by full-matrix least-squares methods on F2 with SHELXL97 [50] program. H atoms on C were geometrically positioned and treated as riding atoms, with C–H = 0.93–0.98 Å, and with Uiso(H) = 1.2Ueq(C). The program Mercury was used for visualization, molecular graphics, and analysis of crystal structures [52]. The software used to prepare material for publication was PLATON [53]. Crystallographic data for the structures in this paper have been deposited with the Cambridge Crystallographic Data Centre as supplementary publication CCDC numbers 1949131 (**8a**), 1949129 (**8b**), 1413575 (**8c**) and 1949130 (**9f**). Copies of the data can be obtained free of charge on application to CCDC, 12 Union Road, Cambridge CB2 1EZ, UK (Fax: +44-01-223-336-033 or E-Mail: deposit@ccdc.cam.ac.uk).

### 3.1. Experimental Section

2-Aminobenzothiazole **1**, CS_2_, iodomethane, glycine, (*l*)-alanine, (*l*)-phenyl-glycine, (*l*)-phenylalanine, (*l*)-valine, (*l*)-leucine, DMF, ethyl alcohol and NaOH were commercial products, which were used as received. Yields, physical appearances, melting points, IR frequencies, and elemental analysis of compound **5a**–**f** and **8a**–**f** are listed in Appendix A. 

### 3.2. General Method for Isothiourea Carboxylates ***5a**–**f***

Amino acid (3.94 mmol), NaOH (3.94 mmol), and 15 mL of ethanol were added into a 100 mL round flask and the mixture was stirred for 2 h at room temperature. Then, compound **2** (1.0 g, 3.94 mmol) was added and the mixture was stirred for additional 96 h at room temperature.

#### 3.2.1. Sodium (*E*)-(3-Benzothiazol-2-Yl-2-Methyl-Isothioureido)-Acetate **5a**

As a general method, starting from 0.295 g of glycine, compound **5a** precipitated from the reaction mixture, ethanol was eliminated, the resulting mixture was cooled to room temperature and 10 mL of acetone were added, the resulting suspension was filtered and washed with cold acetone, obtaining **5a**.3H_2_O as a cream color powder (1.14 g, 82%); mp = 220 °C (dc).

#### 3.2.2. Sodium (*S*,*E*)-(+)-2-(3-Benzothiazol-2-Yl-2-Methyl-Isothioureido)-Propionate **5b**

As a general method, starting from 0.35 g of *l*-alanine, ethanol was evaporated from the homogeneous reaction mixture, the resulting gummy product was dissolved in 10 mL of acetone and filtered, acetone was eliminated and a transparent yellowish ionic liquid compound **5b**.2H_2_O was obtained (1.08 g, 78%), which resulted to be soluble in chloroform.

#### 3.2.3. Sodium (*R*,*E*)-(+)-2-(3-Benzothiazol-2-Yl-2-Methyl-Isothioureido)-Phenyl-Acetate **5c**

As a general method, starting from 0.594 g of *l*-phenylglycine, ethanol was eliminated from the reaction mixture and then suspended in acetone (10 mL). The suspension was filtered and the remaining solid washed with cold acetone and dried to obtain a white powder (1.29 g, 86%).

#### 3.2.4. Sodium (*S*,*E*)-(−)-2-(3-Benzothiazol-2-Yl-2-Methyl-Isothioureido)-3-Phenyl-Propionate **5d**

As a general method, starting from 0.650 g of *l*-phenylalanine, from the same procedure as **5b**, compound **5d** was obtained as brownish liquid (1.26 g, 81%).

#### 3.2.5. Sodium (*S*,*E*)-(−)-2-(3-Benzothiazol-2-Yl-2-Methyl-Isothioureido)-3-Methyl-Butanonate **5e**

As a general method, starting from 0.673 g of *l*-valine, from the same procedure as **5b**, compound **5e** was obtained as clear liquid (1.09 g, 80%).

#### 3.2.6. Sodium (*S*,*E*)-(−)-2-(3-Benzothiazol-2-Yl-2-Methyl-Isothioureido)-4-Methyl-Pentanonate **5f**

As a general method, starting from 0.516 g of *l*-leucine, from the same procedure as **5b** compound **5f** was obtained as yellowish liquid (1.17 g, 81%)

### 3.3. General Method for Isourea Carboxylates ***6a**–**f***

Starting from isothiourea carboxylates **5a**–**f** (3.0 mmol), 10 mL of ethanol and 10 mL of water were added into a 100 mL round flask and the mixture was refluxed for 72 h. For isothiourea **5a**, DMF was used instead of ethanol. The solvent was evaporated from the reaction mixture and the residue suspended in acetone, the suspension was cooled and filtered, the precipitate was washed with acetone and dried. In the hydrolysis of isothioureas **5e**,**f**, the reaction mixture was filtered and the solvents were evaporated. 5 mL of acetone were added and 50 mL of CHCl_3_ were slowly added, the mixture was stirred until a beige solid precipitates, which was filtered and washed with CHCl_3_ to get **6e**,**f**.

#### 3.3.1. Sodium (3-Benzothiazol-2-Yl-Isoureido)-Acetate **6a**

0.32 g, 40% yield; white powder; mp = 210 °C (dc).

#### 3.3.2. Sodium 2-(3-Benzothiazol-2-Yl-Isoureido)-Propionate **6b**

White powder; 0.6 g, 70% yield; mp = 205 °C (dc).

#### 3.3.3. Sodium (3-Benzothiazol-2-Yl-Isoureido)-Phenyl-Acetate **6c**

White powder; 0.75 g, 72% yield; mp = 250 °C (dc).

#### 3.3.4. Sodium 2-(3-Benzothiazol-2-Yl-Isoureido)-3-Phenyl-Propionate **6d**

White powder; 0.81 g, 75% yield; mp = 149–150 °C.

#### 3.3.5. Sodium 2-(3-Benzothiazol-2-Yl-Isoureido)-3-Methyl-Butirate **6e**

White powder; 0.61 g, 65% yield; mp = 199–201 °C.

#### 3.3.6. Sodium 2-(3-Benzothiazol-2-Yl-Isoureido)-4-Methyl-Pentanonate **6f**

White powder; 0.69 g, 70% yield; mp = 201–203 °C.

### 3.4. General Method for SMe-Isothiourea Carboxylate Methyl Esters ***8a**–**f*** and ***9e**,**f*** or Isourea Carboxylate Methyl Esters ***10a**–**f*** and Urea Methyl Ester ***12e**,**f***

In a 100 mL round flask, 3.0 mmol of the corresponding isothiourea carboxylate **5a**–**f** or urea carboxylate **6a**–**f** were dissolved in DMF (10 mL), methyl iodide (3.5 mmol) were added and the mixture was stirred for 12 h on an ice bath and then 12 h at room temperature. At the end of the reaction, 50 mL of water were added and the corresponding ester was extracted with CHCl_3_. Chloroform was eliminated and the *O*-methyl compounds were purified by crystallization in ethanol. Compounds **9e** or **9f** were separated from **8e** or **8f** by crystallization from their ethanol mixture. Compounds **12e** or **12f** were separated from **10e** or **10f** using a chloroform/acetone 10:1 mixture in a silica gel chromatography column. **12e** and **12f** were precipitated from hexane.

#### 3.4.1. (*E*)-2-(3-Benzothiazol-2-Yl-2-Methyl-Isothioureido)-Acetic Acid Methyl Ester **8a**

White crystals; 0.73 g, 83% yield; mp = 145–146 °C.

#### 3.4.2. (*S*,*E*)-(+)-2-(3-Benzothiazol-2-Yl-2-Methyl-Isothioureido)-Propionic Acid Methyl Ester **8b**

White crystals; 0.74 g, 80% yield; mp = 100–101 °C.

#### 3.4.3. (*S*,*E*)-(+)-2-(3-Benzothiazol-2-Yl-2-Methyl-Isothioureido)-Phenyl-Acetic Acid Methyl Ester **8c**

White crystals; 0.83 g, 75% yield; mp = 133–134 °C.

#### 3.4.4. (*S*,*E*)-(−)-2-(3-Benzothiazol-2-Yl-2-Methyl-Isothioureido)-3-Phenyl-Propionic Acid Methyl Ester **8d**

White crystals; 0.64 g, 56% yield; mp = 84–85 °C.

#### 3.4.5. (*S*,*E*)-(−)-2-(3-Benzothiazol-2-Yl-2-Methyl-Isothioureido)-3-Methyl-Butanoic Acid Methyl Ester **8e**

Viscous liquid; 0.49 g, 49% yield.

#### 3.4.6. (*S*,*E*)-(−)-2-(3-Benzothiazol-2-Yl-2-Methyl-Isothioureido)-4-Methyl-Pentanonic Acid Methyl Ester **8f**

Viscous liquid; 0.53 g, 51% yield.

#### 3.4.7. 3-Methyl-2-[2-Methyl-3-(3-Methyl-3*H*-Benzothiazol-2-Ylidene)-Isothioureido]-Butyric Acid Methyl Ester **9e**

White crystals; 0.084 g, 8% yield; mp = 163–164 °C; MS: M + H = 352.1 (79.1%).

#### 3.4.8. 4-Methyl-2-[2-Methyl-3-(3-Methyl-3*H*-Benzothiazol-2-Ylidene)-Isothioureido]-Pentanoic Acid Methyl Ester **9f**

White crystals; 0.27 g, 25% yield; mp = 177–178 °C; MS: M + H = 366.0 (80.1%).

#### 3.4.9. (3-Benzothiazol-2-Yl-Isoureido)-Acetic Acid Methyl Ester **10a**

White powder; 0.65 g, 82% yield; mp = 170–180 °C; MS: M + H = 266.06 (85%).

#### 3.4.10. 2-(3-Benzothiazol-2-Yl-Isoureido)-Propionic Acid Methyl Ester **10b**

White powder; 0.68 g, 86% yield; mp = 149–150 °C; MS: M + H = 280.07 (85.6%).

#### 3.4.11. (3-Benzothiazol-2-Yl-Isoureido)-Phenyl-Acetic Acid Methyl Ester **10c**

White powder; 0.73 g, 72% yield; mp = 112–114 °C; MS: M + H = 342.1 (83.1%).

#### 3.4.12. 2-(3-Benzothiazol-2-Yl-Isoureido)-3-Phenyl-Propionic Acid Methyl Ester **10d**

White powder; 0.32 g, 30% yield, 70–71 °C (mp), MS: M + H = 356.1 (80.6%).

#### 3.4.13. 2-(3-Benzothiazol-2-Yl-Isoureido)-3-Methyl-Butyric Acid Methyl Ester **10e**

Viscous liquid; 0.6 g, 66% yield; MS: M + H = 308.1 (82.7%).

#### 3.4.14. 2-(3-Benzothiazol-2-Yl-Isoureido)-4-Methyl-Pentanoic Acid Methyl Ester **10f**

Viscous liquid; 0.52 g, 54% yield; MS: M + H = 321.0 (84.1%).

#### 3.4.15. 3-Methyl-2-[3-(3-Methyl-3*H*-Benzothiazol-2-Ylidene)-Ureido]-Butyric Acid Methyl Ester **12e**

White powder; 0.23 g, 24% yield; mp = 100–101 °C; MS: M + H = 322.1 (83%).

#### 3.4.16. 4-Methyl-2-[3-(3-Methyl-3*H*-Benzothiazol-2-Ylidene)-Ureido]-Pentanoic Acid Methyl Ester **12f**

White powder; 0.21 g, 21% yield; mp = 87–89 °C; MS: M + H = 336.1 (82%).

### 3.5. General Method for Isourea Amides ***11a**–**f*** or Urea-Amides ***13e**,**f***

In a 100 mL round flask, 3.00 mmol of the corresponding isourea carboxylate methyl esters 10a-f or urea methyl esters **12e**,**f** were dissolved in ethanol (10 mL), methyl amine 40% in water (3.5 mmol) were added and the mixture was refluxed for 24 h. At the end of the reaction, the precipitate was filtered and washed with plenty of acetone.

#### 3.5.1. 2-[3-(3*H*-Benzothiazol-2-Ylidene)-Ureido]-*N*-Methyl-Acetamide **11a**

White powder; 0.63g, 80% yield; mp = 260–270 °C (dc); MS: M + H = 265.1 (82.1%).

#### 3.5.2. 2-[3-(3*H*-Benzothiazol-2-Ylidene)-Ureido]-*N*-Methyl-Propionamide **11b**

White powder; 0.81g, 97% yield; mp = 250–260 °C (dc), MS: M + H = 279.09 (84.8%).

#### 3.5.3. 2-[3-(3*H*-Benzothiazol-2-Ylidene)-Ureido]-*N*-Methyl-2-Phenyl-Acetamide **11c**

White powder; 0.98 g, 96% yield; mp = 270–290 °C (dc); MS: M + H = 341.1 (77.9%).

#### 3.5.4. 2-[3-(3*H*-Benzothiazol-2-Ylidene)-Ureido]-*N*-Methyl-3-Phenyl-Propionamide **11d**

White powder; 1.06 g, 93% yield, mp = 270–280 °C (dc); MS: M + H = 355.1 (77.5%).

#### 3.5.5. 2-[3-(3*H*-Benzothiazol-2-Ylidene)-Ureido]-*N*-Methyl-3-Methyl-Butiramide **11e**

White powder; 0.55g, 60% yield; mp = 205–207 °C (dc), MS: M + H = 307.12 (80.1%).

#### 3.5.6. 2-[3-(3*H*-Benzothiazol-2-Ylidene)-Ureido]-*N*-Methyl-4-Methyl-Pentylamide **11f**

White powder; 0.6 g, 63% yield; mp = 138–140 °C; MS: M + H= 321.1 (76.1%).

#### 3.5.7. 3,*N*-Dimethyl-2-[3-(3-Methyl-3*H*-Benzothiazol-2-Ylidene)-Ureido]-Butiramide **13e**

White powder; 0.2 g, 20% yield; mp = 205–206 °C; MS: M + H = 321.1 (79.1%).

#### 3.5.8. 4-Methyl-2-[3-(3-Methyl-3*H*-Benzothiazol-2-Ylidene)-Ureido]-Pentanoic Acid Methylamide **13f**

White powder; 0.46 g, 46% yield; mp = 210–212 °C, MS: M + H = 335.1 (79.9%).

## 4. Conclusions

A six sodium salt series of isothiourea-carboxylate benzothiazoles **5a**–**f**, as well as their methyl ester derivatives **8a**–**f**, were obtained in moderate to good yields by the reaction of dimethylcarbonimidate benzothiazole **2** with sodium salts of glycine, (*l*)-alanine, (*l*)-phenylglycine, (*l*)-phenylalanine, (*l*)-valine, and (*l*)-leucine in stirring ethanol at room temperature and further methylation under mild conditions. The reaction is stereo selective, only the *E*-isomer was isolated, the X-ray structure of (*R*,*E*)-methyl-2-((benzothiazol-2-ylimino)(methyl-thio)methylamino)-2-phenylacetate 8c confirmed the stereochemistry of the reaction. The structures of **8a** and **8c** are stabilized by three center hydrogen bonding interactions N3···H12···O14 between the amino N12H12 with the nitrogen atom of benzothiazole ring and the oxygen atom of the carbonyl group, forming two intramolecular adjacent *S*(*6*) and *S*(*5*) rings, respectively. This finding suggests the stereochemical assistance of the reaction by hydrogen bonding. When the same reactions were carried out in the presence of water, the urea-carboxylate benzotiazoles **6a**–**f** were obtained. Their further methylation produced the corresponding methyl esters **10a**–**f**. In the methylation reaction of sodium isothiourea-carboxylates **5e**,**f** and urea-carboxylates **6e**,**f**, the corresponding N3Me methyl esters **9e**,**f** and **12e**,**f** were produced as byproducts, which were isolated. Methyl esters **8a**–**f** or **10a**–**f** and **9e**,**f** or **12e**,**f** were used as starting materials to produce the corresponding urea carboximides **11a**–**f** and **13e**,**f** by the reaction with methyl amine. Further studies on the synthesis of chiral guanidines from SMe-isothioureas **8** are in progress.

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
