# Peer review of "Isothioureas, Ureas, and Their N-Methyl Amides from 2-Aminobenzothiazole and Chiral Amino Acids"

_molecules, 2019, doi:10.3390/molecules24183391_

Round 1

Reviewer 1 Report

This manuscript describes the reaction of 2-dithiomethylcarboimidatebenzothiazole with a series of six chiral amino-acids. A variety of isothioureas, ureas and their N-methylamides were obtained. These heterocycles are important structures in biological and pharmaceutical fields. The synthetic method is simple. The proposed reaction mechanism is plausible. Products are high yields and their structure are well characterized. From these reasons, I recommend this manuscript for publication, provided that the following issues are addressed.

Were all products isolated as optically pure ones without racemization? Line 183; fig 2 Scheme 3 8a-f; SMe not SCH3 It is better to show products yields in Tables.

Reviewer 2 Report

Manuscript Number: molecules-585333

entitled: Isothioureas, Ureas and their N-methylamides From 2-Aminobenzothiazole and Chiral Aminoacids

It’s an interesting paper. The present version is very hard to follow. I have the following question/comments to the authors.

Synthesis. Step 1-st (synthesis of products 2 is already known

(Cruz, Alejandro; Padilla-Martinez, Itzia I.; Garcia-Baez, Efren V.Molecules2012vol. 17# 9p. 10178 - 10191,14).

Step 2-nd is already known

(a. Tellez, Fabiola; Cruz, Alejandro; Lopez-Sandoval, Horacio; Ramos-Garcia, Iris; Gayosso, Martha; Nely Castillo-Sierra; Paz-Michel, Brenda; Noeth, Heinrich; Flores-Parra, Angelina; Contreras, RosalindaEuropean Journal of Organic Chemistry2004# 20p. 4203 – 4214, b. Cruz, Alejandro; Padilla-Martinez, Itzia I.; Garcia-Baez, Efren V.Molecules2012vol. 17# 9p. 10178 - 10191,14).

The last step is already known

(Cruz, Alejandro; Padilla-Martinez, Itzia I.; Garcia-Baez, Efren V.Molecules2012vol. 17# 9p. 10178 - 10191,14).

So what is really new? There is no clear information about what was already made by authors, but should be (see line 58).

English Line 78 “l-phenylalanine (R = Bn, d), l-valine (R = iPr, e) and l-leucine (R = iBu, f) which are stable as switterions” or line 175 “solution of compound 10c (R = Ph), they appear at 10.6 (broad) and 7.8 ppm (doublette” Please add whole NMR characteristic to experimental part or whole experimental part add to sm? Please add coupling constants e.g. 3JH.H = 7Hz, and MS and HRMS data for all new compounds. For some protons hydrogen-deuterium (H/D) exchange is possible e.g. if they are acidic, in this case there is acid - base exchange. So the sentence “show three NH deuterium labile hydrogen atoms” should be change, it’s no a true. X-ray data to sm, only the most significant data in main text. Please unify reaction procedures e.g. 307 with cold acetone and dried to obtain 29 g (86%) as a white powder.. to (1.29 g, 86%). Please add instrumentation to experimental part. Please add whole characteristic 1H, 13C NMR, MS for results to the experimental part. Line 194 „In compounds 13e,f the urea NH appears as a doublet at 5.9, 5.7 ppm (3J = 9.3, 8.2 Hz)” it is a mistake, single doublet with 2 coupling constants? No, it is a mistake. Line 195 “amide NH appears as a quartet at 6.6, 6.4 ppm ( 3J = 4.4, 4.7 Hz ).” it is a mistake, maybe double doublet “dd” Line 344 “separated from a CHCl3/H2O (NaCl) system.” What does mean? Exp. Part must be re-write, and all necessary data add.

In my judgment, there is good publishable science in this manuscript, but it needs some work before it can be accepted.

Round 2

Reviewer 2 Report

Dear Authors,

please change only MS data M+1 to M+H e.g. line 348

Author Response

M+1 were changed to M+H